# Survey of Danish Head and Neck Cancer Patients’ Positions on Personalized Medicine, Gene Tests, and Personalized Follow-Up

**DOI:** 10.3390/jpm14040404

**Published:** 2024-04-11

**Authors:** Christian Sander Danstrup, Maria Andersen, Søren Lundbye-Christensen, Mia Sommer, Nina Munk Lyhne

**Affiliations:** 1Department of Otorhinolaryngology-Head and Neck Surgery, Aalborg University Hospital, 9000 Aalborg, Denmark; 2Department of Clinical Medicine, Aalborg University, 9000 Aalborg, Denmark; 3Clinical Cancer Research Center, Aalborg University Hospital, 9000 Aalborg, Denmark; 4Department of Clinical Oncology, Aalborg University Hospital, 9000 Aalborg, Denmark; 5Research Data and Biostatistics, Aalborg University Hospital, 9000 Aalborg, Denmark; 6Department of Hematology, Aalborg University Hospital, 9000 Aalborg, Denmark; 7School of Nursing, University College Northern Denmark, 9000 Aalborg, Denmark

**Keywords:** head and neck neoplasms, personalized medicine, otorhinolaryngology, shared decision making

## Abstract

The field of personalized medicine (PM) has grown rapidly because of the “omics revolution”, but PM may be difficult for patients to comprehend. This study sought to explore head and neck cancer (HNC) patients’ positions and knowledge of PM, gene tests, and follow-up and to compare HNC patients’ positions to a sample from a national Danish questionnaire. To do this, patients with prior HNC were invited to participate in a questionnaire. Initial interviews revealed a heterogenic understanding of PM between patients. A total of 226 patients were included in the survey and 177 patients with complete data were included for analysis. Most patients were more positive than negative towards gene tests and gene research (83% and 93%, respectively), but 72% had little or no knowledge of the subject. Almost all patients, 98%, were satisfied with their follow-up. Significantly more patients with HNC were positive towards gene research compared to a sample from a national Danish questionnaire (*p* < 0.001). Patients with HNC were positive towards gene tests and PM, but patients may not understand or comprehend the information given, and it is important to inform and educate patients and health professionals to establish common ground in PM.

## 1. Introduction

The laboratory evolution has created a plethora of new diagnostic and prognostic tests for cancer. What once was expensive and time-consuming testing can now be performed fast and inexpensively [1]. The availability of new technologies to potentially stratify patients and treatments has introduced new terms such as “Precision Medicine” and “Personalized Medicine” (PM) in contrast to the “one-size-fits-all” principle [2].

Following treatment, cancer survivors face a life with risk and fear of recurrence. Different follow-up regimens exist, depending on the type of cancer, and include a variety of clinical and biochemical evaluations and diagnostic images.

The patients’ positions on follow-up vary between studies and cancer types because some patients find the current number of follow-up visits too frequent, while others prefer frequent in-office visits [3,4].

Many publications describe the promising potential of new laboratory tests and PM, but few studies report the patients’ perspectives and positions on PM. In general, cancer patients and the general public are positively inclined towards PM but may have limited knowledge [5,6,7].

Head and neck cancer (HNC) most commonly refers to squamous cell carcinomas (SCC) arising from the mucosal surfaces of the head and neck (oral cavity, pharynx, and larynx), but according to the Union for International Cancer Control (UICC) classification of malignant tumors, head and neck tumors also include malignancies in the major salivary glands and thyroid gland [8]. The incidence of HNC is rising and when including all subsites, more than 1,500,000 new cases are found worldwide each year [9]. The most common risk factors in SCC of the head and neck have been tobacco and alcohol, but within the last decades, the number of human papillomavirus (HPV) associated tumors has grown, especially in the oropharynx [10,11]. The risk factors for thyroid cancer are more heterogeneous and include environmental exposure, sex, and family history [12].

Socio-economic position and social inequality are often mentioned as important risk factors for cancer, and low education has been associated with a higher incidence of HNC [13]. Due to different risk factors and epidemiology of the various subsites in HNC, certain groups may have different demography and social status compared to others (e.g., patients with thyroid and salivary gland tumors compared to patients with SCC). Disparities are also found within the SCC groups as patients with HPV-positive oropharyngeal tumors have been shown to have longer education compared to patients with HPV-negative tumors [14].

The treatment of HNC depends on the anatomical subsite and histology, but usually consists of surgery, radiotherapy with or without chemotherapy, or a combination thereof. A large proportion of the patients, especially SCC patients presenting with late-stage disease at diagnosis, make treatment with curative intent difficult [15,16]. Even in smaller tumors, the surgical and oncological treatment leaves the patients with significant morbidity and treatment sequelae. Therefore, it is necessary to be able to diagnose at an earlier stage and tailor the treatment to the individual. Due to the laboratory evolution, researchers now have a better understanding of the tumors and their microenvironment, and clinical trials are exploring the use of extensive tumor evaluation in HNC and other tumors [17,18]. Recently, immunotherapy with a programmed cell death-ligand 1 (PD-L1) inhibitor, has been approved as a treatment option in selected patients with recurrent SCC [19,20]. Collectively, PM is slowly but steadily finding its place in the treatment of HNC, and patients may be asked to make decisions regarding their treatment based on information from complex and extensive laboratory evaluations. 

Knowing the patient’s position and knowledge regarding their disease, treatment, and follow-up is central to securing a therapeutic alliance. It should be addressed because the patient’s knowledge may affect their choice of treatment [21]. Furthermore, knowing the patient’s position on follow-up also enables a potential tailoring of individual follow-up plans.

The primary aim of this study was to evaluate HNC patients’ positions and knowledge regarding PM and gene tests (GT). The secondary aims were to evaluate HNC patients’ positions on personalized follow-up and to compare HNC patients’ positions to the sample from the national Danish questionnaire.

## 2. Materials and Methods

### 2.1. Design

The study was a cross-sectional study using a structured questionnaire reported following the Checklist for Reporting of Survey Studies (CROSS) (Appendix A) [22]. The study was completed at the Department of Otorhinolaryngology–Head and Neck Surgery, Aalborg University Hospital. The hospital has a catchment area of 600,000 inhabitants. All patients treated for HNC are offered follow-up according to Danish guidelines [23].

The questionnaire was validated by semi-structured cognitive interviews to investigate future respondents’ understanding. Following recommendations by Beaton et al., a total of 30 respondents with prior HNC who were actively monitored were invited to fill out the questionnaires and answer predefined probes as described by Beatty et al. (Appendix A) [24,25]. Details of the pre-tested respondents can be found in Table 1.

Following the cognitive interviews, the authors revised and proofread the questionnaire. Patients with prior HNC (including salivary and thyroid glands) who were actively monitored were invited to participate in the study during a planned follow-up visit. Involved physicians were informed of the study prior to commencement. After the consultation, the patients were asked to fill out the questionnaire on paper unsupervised in the waiting room and return it to the reception. Patients were included from 1 May 2021 to 31 December 2021. We aimed to include 250 patients, but due to structural challenges in the outpatient clinic, the study had to be closed before the planned number of patients was reached.

#### 2.1.1. Covariates

The physicians registered clinical covariates such as tumor type and extension. The patients were asked to register covariates such as age and gender. The full list of covariates can be found in Table 2. 

#### 2.1.2. Observations

The questionnaire in Danish had a Likert-scale design with two parts:

Part One:

In 2016, a survey was developed on behalf of the Danish Ministry of Health and Danish Regions. The survey was designed to evaluate the general Danish populations’ opinion on PM, GT, gene research (GR), and how the Ministry of Health could inform the general Danish population about PM and GT. In addition to the questionnaire, the survey consisted of two public panels of citizens from the general Danish population. We used the part of the questionnaire regarding interest and knowledge of PM, GT, and GR [5].

Part Two:

The authors developed Part Two of the questionnaire with inspiration from Kothari et al. regarding the HNC patients’ positions on follow-up [4].

The Danish questionnaire and a non-validated English translation can be found in Appendix A.

### 2.2. Ethics and Approvals

The study was approved by The Danish Data Protection Agency of the Northern Region of Denmark (ID:2020-089).

Under Danish law, approval by the Committee on Health Research Ethics is not required for this type of study. The study was conducted according to the Helsinki Declaration and participants provided oral and written consent. 

Questionnaires were handed out on paper, and study data was collected and managed using Research Electronic Data Capture (REDCap) hosted by the North Denmark Region. REDCap is a secure, web-based software platform designed to support data capture for research studies, providing (1) an intuitive interface for validated data capture; (2) audit trails for tracking data manipulation and export procedures; (3) automated export procedures for seamless data downloads to common statistical packages; and (4) procedures for data integration and interoperability with external sources [26,27].

### 2.3. Statistics

Missing data was explored. Of the 226 included, 78% had complete data. In general, missing data was few, and no variable had more than 5% missing data (Table 2). Cases with complete data and cases with missing data were compared (Fisher’s Exact and unpaired t-test) to examine potential differences between the two groups. Due to statistical differences between groups in one covariate, multiple imputation using chained equations (MICE) were performed to examine the potential impact on the results [28]. Analyses were performed on complete as well as imputed data and the results were compared to examine potential statistical differences if using cases with complete data or cases with imputed data.

Answer proportions between the sample from the national Danish questionnaire and HNC patients were compared with Fisher’s exact test [5].

Fisher’s exact test, one-way ANOVA, and linear regression were used to explore potential associations between covariates and outcomes. For the comparison analyses, the Likert scale was categorized into three groups (1–3 = positive, 4 = neutral, 5–7 = negative).

The data was analyzed using Stata statistical software (version 16, StataCorp LP, College Station, US). Statistical significance was defined as *p* ≤ 0.05.

## 3. Results

### 3.1. Cognitive Interviews

Of the 30 invited participants, one patient did not finish the interview and was excluded. The remaining 29 patients were positive towards GT and GR, but the probe question “After having read the introduction, tell me, with your own words, your understanding of the concept PM” revealed a large variation in answers. A total of 18/29 (62%) respondents perceived that PM was their own medicine (such as regular prescription medicine and nutritional supplements). The remaining 11 respondents answered that PM was tailored treatment on an individual level or variations.

### 3.2. Missing Data

Fisher’s exact test revealed no statistical difference in sex, education, or municipality between cases with complete data and cases with missing data. An unpaired *t*-test showed that cases with missing data were statistically significantly older (*p* = 0.0008) than cases with complete data. The details of missing data can be found in Table 2.

All analyses regarding patients’ positions on GT, follow-up plans, and comparisons to the sample from the general Danish population were performed on complete cases as well as imputed data. Comparisons between data from complete cases and imputed data. did not reveal significant differences. Therefore, only analyses of complete cases are presented.

### 3.3. Demographics

A total of 226 patients participated. Of these, 49 patients had missing data, leaving 177 complete cases for further analysis. Of the complete cases, 113 (64%) were male, and the mean age was 59.6 years. The details on patient demographics are presented in Table 3. The tumor type, stage, and follow-up status are presented in Table 4.

### 3.4. The Patients’ Positions on Gene Tests

A total of 147/177 (83%) patients were more positive than negative towards GT. No statistically significant difference in position depending on age (*p* = 0.593), sex (*p* = 0.448), municipality (*p* = 0.744), education (*p* = 0.594), and tumor type (*p* = 0.794) were found. Of the 177 included, 165 (93%) patients found GR in Denmark important. Despite the general positivity towards GT and GR, as many as 127/177 (72%) patients answered that they knew nothing or little about GT. 

Of the included patients, only five (3%) patients had had a GT performed, three in relation to medical evaluation or treatment at the hospital, one on their own initiative as well as in relation to medical evaluation, and one on their own initiative only. An additional six patients had been offered a GT but had declined. When presented with the question of whether they would prefer to be informed if a GT as a part of a research project incidentally had revealed potential consequences regarding their health, 104 (59%) patients answered “yes, no matter what”, 42 (24%) answered “yes, but only if a treatment option exists”, 13 (7%) answered “no”, and 18 (10%) patients were unsure.

### 3.5. Positions on Current Follow-Up Plans and Potential New Follow-Up Methods

When asked, 174/177 (98%) patients stated that they were satisfied with the follow-up plan that they had followed or that was planned for them. Two patients were not satisfied, and one was unsure.

Given the chance to suggest changes, 146/177 (82%) patients stated that they would not change anything. Nine (5%) patients proposed more scans, four (2%) patients proposed more controls (*n* = 4), and 10 (6%) patients proposed more scans and more controls.

One patient stated either less or no controls but with the possibility to contact the department. One stated more scans and less control but with the possibility to contact the department. Finally, only one stated fewer controls.

After answering their position on the current follow-up plans, the participants were asked their position on follow-up based on classic outpatient visits compared to blood or salivary samples, given that the modalities were equal regarding diagnosing recurrence. A total of 91/177 (51%) were negative towards the laboratory diagnostics, 36/177 (20%) patients were neutral, and 50/177 (28%) were positive. When questioned if they would accept any uncertainty in such tests given, they could be performed at home or by their general practitioner, 85/177 (48%) stated that they would not accept any sort of uncertainty, 19/177 (11%) would accept an uncertainty of different degrees (range 1–20%), and 73/177 (41%) stated that they never would choose the blood or salivary test.

When given a question regarding the potential prognostic capabilities of molecular cancer diagnostics, 131/177 (74%) answered that they would know this prognosis, 6/177 (3%) did not want this information, and 40/177 (23%) were unsure.

When questioned, 73/177 (41%) had received answers from laboratory analyses or diagnostic imaging via telephone or secure digital communication. Of these, 62/73 (85%) were planned answers, but 15% experienced being contacted due to unexpected findings. When given the choice, 113/177 (64%) patients stated that they would never choose to receive answers from the hospital outside the outpatient clinic. The municipality (distance needed to travel to the hospital) did not affect the patients’ choice statistically significantly (*p* = 0.199). The longest potential distance for patients in our study was 135 km. The bar charts showing the distributions of the answers from the questionnaire can be found in Appendix A.

### 3.6. Comparisons between the Head and Neck Cancer Patients from the Present Study and Respondents from the Danish National Questionnaire on Personalized Medicine

We examined similarities and discrepancies between the HNC group and the data from the Danish national questionnaire [5]. The HNC patients were more positive towards GT and GR but had significantly less PM knowledge than the respondents from the national questionnaire (Table 5). Respondents from the national questionnaire had significantly higher levels of education than the HNC patients; instead, the HNC patients in our study had more resemblance to the overall educational levels within the general Danish population.

## 4. Discussion

In general, the patients in our study were positive towards GT and GR, even though they stated little or no knowledge of the subject. No covariate was associated with a specific patient position on PM.

In our study, 98% of the patients were satisfied with their follow-up program, and 82% stated that they would not change anything, given the opportunity. This result contrasts with the study by Kothari et al., where 84% of patients with HNC felt that their head and neck follow-up visits were too frequent, and 73% of patients were in favor of less intensive follow-up plans with the possibility of requesting appointments, based on problems [4].

In our study, the majority were satisfied with the follow-up plan. Only a few patients proposed more scans and/or controls. Our results align with other studies across different cancer types [29,30,31]. Fidjeland et al. found that patients with gynecological cancer would rather have follow-up visits at university hospitals than with their general practitioners. However, they found a difference in willingness depending on whether the patients had started follow-up or not [29]. In our study, all patients were actively monitored. This difference may explain the trend because patients seek continuity and tend to prefer existing plans/status quo [32]. Our results may have been different if the patients had been asked before enrollment in their follow-up plan.

Our study found that many of the HNC patients had received answers from laboratory analyses or diagnostic images via telephone or secure digital communication, but when asked, the majority in fact would never choose to receive answers without being physically present in the outpatient clinic. This preference was not affected by the distance needed to travel. Compared to other countries, the distances needed to travel in Denmark may be short. In larger countries, patients may have chosen differently if they were to travel further. We did not explore the reasons why the patients were negative towards laboratory samples at home or at their general practitioner further, nor their reasons for preferring follow-up in the outpatient clinic. It would be interesting to explore our patients’ positions on telemedicine (if all technical barriers such as computers and sufficient internet connection had been addressed). Following the COVID-19 pandemic, telemedicine has gained more interest, and studies from the United States, Australia, and the Netherlands have also proven that HNC patients are positive towards telemedicine as a part of the outpatient monitoring, but the lack of physical evaluation remains a limitation [33,34].

Compared to the sample from the national Danish questionnaire, the patients in our study were older and more frequently male. We found statistically significant differences in the proportions of patients with HNC who were positive towards GR compared to the sample from the national Danish questionnaire. This result may be because the HNC patients were more aware of the potential consequences of new diagnostic tests because of their prior cancer and active monitoring. Sommer et al. observed outcomes similar to ours in a cohort of patients with hematological cancers, who were more positive towards PM but had lower knowledge levels than the sample from the national Danish questionnaire [35]. A supplementary survey performed in 2019 found that the majority (75%) of a new sample of the general Danish population still found gene research important [36]. Because the aims and questionnaire designs were different, comparisons of the data to our outcome were not possible. The national questionnaires were completed in 2016 and 2019, and our study was completed in 2021. Knowledge and opinions regarding GT and GR may have changed in the time between the questionnaires. The proportion of patients in our study who stated little or no knowledge of GT was significantly larger than in the sample from the national Danish questionnaire. One reason for this result may be the differences in age and education between the two groups because the proportion of people with higher education was significantly higher in the sample from the national Danish questionnaire. 

It is noteworthy that the educational distribution of the respondents in the national Danish questionnaire differs from the general Danish population. A reason for this may be that people with higher education are more interested in new research and are more willing to participate in such surveys. HNC patients are usually reported to have a lower educational level compared to the general public and other tumors [13,14]. However, the HNC patients in our study compared more to the general Danish population, which may reflect the above-mentioned greater interests in participating in survey studies among persons with higher educations. 

The cognitive interviews revealed heterogenicity in understanding the concept of PM. When implementing new treatment or follow-up plans, exploring the knowledge among the recipients and addressing this issue when planning PM and follow-up may be beneficial. Differences in health literacy among patients are well known and methods to evaluate and overcome genetic literacy deficits have been described [37,38]. Studies have shown that targeted consumer-level information and genetic counseling educational tools can enhance the patients’ understanding of PM [39,40]. A combination of targeted patient information and the standardized education of health professionals may help with shared decision-making on PM, and not only HNC patients, but patients across all medical specialties.

### Strengths and Limitations

The study was undertaken at the only oncological center in the North Region of Denmark, and the population-based setting ensured that the invited cohort represents the Danish HNC population. However, the study also has limitations. The included cohort might be biased by selection because we do not know how many patients declined to participate or their reasons. Further, the heterogenicity of the understanding of PM is also a weakness. Interviews with all patients could have secured a more homogeneous understanding of PM, but this opportunity was not possible in the clinical setting.

Due to the early termination of the study, considerations regarding sample size should be made, especially when it comes to the comparisons between HNC patients and the sample from the national Danish questionnaire in Table 5. A smaller sample size increases the risk of type 2 errors, and post hoc analyses showed that we had to have a 9% difference between groups to detect a statistically significant difference. In the question regarding patients’ positions on GT, there was a 6% difference between groups. A larger sample size may have changed the results in either direction. However, when looking at the raw data from the GT and GR questions from our questionnaire, the answers are so skewed towards being positive, that even if we had reached the intended number of patients and the remaining had answered negatively, the group would still be more positive than negative. We therefore believe that our conclusions are valid despite incomplete inclusion into the study.

## 5. Conclusions

Patients with HNC are positive toward PM but have limited knowledge. Currently, patients still prefer follow-ups in the outpatient clinic.

The patients’ perceptions and positions on PM should be explored further, as should the information and educational tools to ensure adequate information for patients and health professionals.

## Figures and Tables

**Table 1 jpm-14-00404-t001:** Patient demographics and tumor types of the 29 head and neck cancer patients who were interviewed during the evaluation process and pre-testing of the questionnaire.

Number of Patients	29 ^1^	
Age, years, mean, (SD ^2^)	66	(10.4)
Sex, *n* (%)		
Male	17	(59)
Female	12	(41)
Municipality, *n* (%)		
Hjørring	3	(10)
Brønderslev	1	(3)
Frederikshavn	1	(3)
Jammerbugt	3	(10)
Mariager Fjord	1	(3)
Morsø	1	(3)
Rebild	1	(3)
Thisted	2	(7)
Vesthimmerland	3	(10)
Aalborg	13	(45)
Level of education (*n*, %)		
Primary education	5	(17)
General upper secondary education	2	(7)
Vocational education	12	(41)
Short-term higher education < 3 years	5	(17)
Medium-term higher education 3–4 years	4	(13)
Long-term higher education ≥ 5 years	1	(3)
Research education (Ph.D.)	0	(0)
Tumor site, *n* (%)		
Oral cavity	10	(34)
Rhinopharynx	0	(0)
Oropharynx	6	(20)
Hypopharynx	1	(3)
Larynx	4	(14)
Salivary glands	3	(10)
Thyroid gland ^3^	3	(10)
Sino-nasal	1	(3)
Unknown primary	1	(3)

^1^ A total of 30 patients were invited, but one patient did not finish the interview and was excluded, ^2^ Standard Deviation, ^3^ Excluding thyroid microcarcinomas.

**Table 2 jpm-14-00404-t002:** Full list of covariates and the distribution of missing data among the total 226 included patients.

Covariate	Missing, *n* (%)	
Tumor type	0	(0)
Tumor-stage	0	(0)
Nodal-stage	0	(0)
Metastasis-stage	0	(0)
Follow-up time	0	(0)
Treatment	0	(0)
Treatment status	0	(0)
Sex	0	(0)
Age	0	(0)
Municipality	8	(4)
Education	7	(3)
Position on PM ^1^	5	(2)
PM ^1^ knowledge	7	(3)
Have had gene test performed	11	(5)
Position on PM ^1^ research	3	(1)
Willing to know consequence of gene test	2	(1)
Satisfied with follow-up	3	(1)
Change in follow-up	6	(3)
PM ^1^ or regular follow-up	3	(1)
Risk willingness if hypothetically choosing PM ^1^ over regular follow-up	6	(3)
Wanting to know prognosis if available via PM ^1^	6	(3)
Had received results from the hospital via telephone during their current follow-up plan	11	(5)
Risk willingness when hypothetically choosing to receive results via telephone	10	(4)

^1^ Personalized medicine.

**Table 3 jpm-14-00404-t003:** Demographics of head and neck cancer patients with complete questionnaires.

	Total	Tumors from Mucosal Surfaces ^1^	Thyroid and Salivary Glands
Number of patients, *n* (%)	177	114	63
Age, years, mean (SD ^2^)	59.6 (13.1)	63.6 (10.0)	52.4 (14.9)
Sex, *n* (%)			
Male	113 (64)	90 (79)	23 (37)
Female	64 (36)	24 (21)	40 (63)
Municipality, *n* (%)			
Hjørring	23 (13)	13 (11)	10 (16)
Brønderslev	10 (6)	9 (8)	1 (2)
Frederikshavn	15 (8)	9 (8)	6 (10)
Jammerbugt	13 (7)	8 (7)	5 (8)
Mariager Fjord	17 (10)	10 (9)	7 (11)
Morsø	3 (2)	1 (1)	2 (3)
Rebild	11 (6)	6 (5)	5 (8)
Thisted	8 (8)	6 (5)	2 (3)
Vesthimmerland	8 (8)	2 (2)	6 (10)
Aalborg	69 (39)	50 (44)	19 (30)
Level of education *n* (%)			
Primary education	36 (20)	29 (25)	7 (11)
General upper secondary education	10 (6)	7 (6)	3 (5)
Vocational education	61 (34)	39 (34)	22 (35)
Short-term higher education < 3 years	20 (11)	10 (9)	10 (16)
Medium-term higher education 3–4 years	34 (19)	21 (18.4)	13 (21)
Long-term higher education ≥ 5 years	15 (8)	8(7)	7 (11)
Research education (Ph.D.)	1 (1)	-	1 (2)

^1^ Tumors from the oral cavity, pharynx, larynx, sino-nasal, and unknown primary combined, ^2^ Standard deviation.

**Table 4 jpm-14-00404-t004:** Tumor type, stage, and follow-up status of patients with head and neck cancer with complete questionnaires.

Number of Patients	177	
Tumor site, *n* (%)		
Oral cavity	17	(10)
Rhinopharynx	5	(3)
Oropharynx	63	(36)
Hypopharynx	5	(3)
Larynx	15	(8)
Salivary glands	15	(8)
Thyroid gland ^1^	48	(27)
Sino-nasal	3	(2)
Unknown primary	6	(3)
T stage, *n* (%)		
T1	56	(32)
T2	64	(36)
T3	39	(22)
T4	12	(7)
T0	6	(3)
N stage, *n* (%)		
N0	89	(50)
N1	46	(26)
N2	35	(20)
N3	7	(4)
M stage, *n* (%)		
M0	177	(100)

^1^ Excluding thyroid microcarcinomas.

**Table 5 jpm-14-00404-t005:** Comparisons of answer proportions regarding gene tests and research between the sample from the Danish national questionnaire and patients with head and neck cancer.

	Danish Population ^1^ *n* = 1005	HNC ^2^ *n* = 177	*p*
Males (%)	44	64	<0.001 ^3^
Age, mean, (SD)			
Male	57.2 (15.0)	63.2 (10.8)	<0.001 ^4^
Female	49.8 (15.8)	53.3 (14.3)	0.069 ^4^
Find gene test more interesting than worrying (%)	77	83	0.077 ^3^
Have little or no knowledge regarding gene tests (%)	55	72	<0.001 ^3^
Are positive towards genetic research (%)	83	93	<0.001 ^3^
Higher education ^5^ (%)	63	40	<0.001 ^3^

^1^ Sample from the general population from the Danish questionnaire in 2016 [5]. ^2^ Head and neck cancer patients from Aalborg University Hospital. ^3^ Fisher’s exact test. ^4^ Unpaired *t*-test. ^5^ All categories with higher education and Ph.D. from Table 1 combined.

## Data Availability

Data, survey, and approach are available in Appendix A. Additional data can be requested from the corresponding author and will be provided in accordance with Danish laws and regulations. Note that according to Danish law, individual-level data may not be deposited.

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
