# Peer review of "Survey of Danish Head and Neck Cancer Patients’ Positions on Personalized Medicine, Gene Tests, and Personalized Follow-Up"

_jpm, 2024, doi:10.3390/jpm14040404_

Round 1
Reviewer 1 Report
Comments and Suggestions for Authors
The authors present a survey on the perception of personalized medicine among the head and neck cancer patients.
The study is certainly well conducted and does not present serious flaws, minor limitations are also discussed.
I have no major remarks to do.
My observations follow:
- Line 223-224 “In our study, the majority were satisfied with the follow-up plan or proposed more scans and/or controls.” This affirmation is somewhat confusing. Regarding what is reported in lines 181-183, the majority of patients did not suggest any changes in follow-up. Only a minor part of them suggested an increase in scans, checks or both.
- Table 3 shows that only 9% of participants had a long-term higher education. Is this representative of the Danish population or not? Maybe it would be appropriate to discuss this aspect.
- The demographics do not include any information about the patients' relatives or friends' experience with PM. Is it possible that a part of participants had previous indirect experience with it and would be more prone to accept it?
- The authors clearly stated the limits of the study. In my opinion, one of the most important factors is the variability of the questionnaire administration modality in a low-educated population. Could the assistance during questionnaire compilation be standardized to reduce misinterpretation of the questions?
Reviewer 2 Report
Comments and Suggestions for Authors
The authors have presented an article on "Survey of Danish Head and Neck Cancer Patients’ Positions on 2 Personalized Medicine, Gene Tests, and Personalized Follow Up". The study work indeed fit into the scope of the journal. The manuscript is well written and easy to understand.
(1) Suggest to describe HNC more in the introduction. Discuss as well why this particular cancer is selected.
(2) Table 1 title "Patient demographics and tumor types of the 29 head and neck cancer patients". Why only 29 patients? Was this the pilot study process? If yes, please mention "pilot study".
(3) Suggest to use more updated references to discuss in Introduction and Discussion part.
